# Screening and Stability Evaluation of Freeze-Dried Protective Agents for a Live Recombinant Pseudorabies Virus Vaccine

**DOI:** 10.3390/vaccines12010065

**Published:** 2024-01-09

**Authors:** Yan Liu, Suling Zhang, Shuai Wang, Chunhui Zhang, Xiaorui Su, Linghua Guo, Xiaofei Bai, Yuxin Huang, Wenqiang Pang, Feifei Tan, Kegong Tian

**Affiliations:** 1National Research Center for Veterinary Medicine, Luoyang 471000, China; liuyan@pulike.com.cn (Y.L.); zhangsuling@pulike.com.cn (S.Z.); wangshuai@pulike.com.cn (S.W.); zhangchunhui@pulike.com.cn (C.Z.); suxiaorui@pulike.com.cn (X.S.); guolinghua@pulike.com.cn (L.G.); baixiaofei@pulike.com.cn (X.B.); huangyuxin@pulike.com.cn (Y.H.); pangwenqiang@pulike.com.cn (W.P.); 2Pulike Biological Engineering Inc., Luoyang 471000, China; 3College of Veterinary Medicine, Northwest A&F University, Yangling 712100, China

**Keywords:** PRV, lyoprotectant, freeze-drying, thermal stability, Arrhenius equation

## Abstract

Infection of pigs with the pseudorabies virus (PRV) causes significant economic losses in the pig industry. Immunization with live vaccines is a crucial aspect in the prevention of pseudorabies in swine. The TK/gE/gI/11k/28k deleted pseudorabies vaccine is a promising alternative for the eradication of epidemic pseudorabies mutant strains. This study optimized the lyophilization of a heat-resistant PRV vaccine to enhance the quality of a live vaccine against the recombinant PRV rHN1201^TK−/gE−/gI−/11k−/28k−^. The A4 freeze-dried protective formulation against PRV was developed by comparing the reduction in virus titer after lyophilization and after seven days of storage at 37 °C. The formulation contains 1% gelatin, 5% trehalose, 0.5% poly-vinylpyrimidine (PVP), 0.5% thiourea, and 1% sorbitol. The A4 freeze-dried vaccine demonstrated superior protection and thermal stability. It experienced a freeze-dried loss of 0.31 Lg post-freeze-drying and a heat loss of 0.42 Lg after being stored at a temperature of 37 °C for 7 consecutive days. The A4 freeze-dried vaccine was characterized through XRD, FTIR, and SEM analyses, which showed that it possessed an amorphous structure with a consistent porous interior. The trehalose component of the vaccine formed stable hydrogen bonds with the virus. Long-term and accelerated stability studies were also conducted. The A4 vaccine maintained viral titer losses of less than 1.0 Lg when exposed to 25 °C for 90 days, 37 °C for 28 days, and 45 °C for 7 days. The A4 vaccine had a titer loss of 0.3 Lg after storage at 2–8 °C for 24 months, and a predicted shelf life of 6.61 years at 2–8 °C using the Arrhenius equation. The A4 freeze-dried vaccine elicited no side effects when used to immunize piglets and produced specific antibodies. This study provides theoretical references and technical support to improve the thermal stability of recombinant PRV rHN1201^TK−/gE−/gI−/11k−/28k−^ vaccines.

## 1. Introduction

Pseudorabies (PR) is a highly infectious, acute, febrile, contagious disease in swine caused by PRV. The primary clinical features of PR include high mortality encephalitis in newborn piglets, abortion and stillbirth in sows, as well as respiratory diseases and delayed growth in fattening pigs [1]. Since it was first discovered in Hungary in 1902, PRV has continued to cause significant economic losses around the world [2,3].

Currently, the most effective measure for the prevention and control of PRV is vaccination [4]. The PRV gE-deleted vaccine, which has been widely used around the world, provides reliable protection to immunized pigs. The pseudorabies virus has been eradicated from domestic pigs in North America and Europe [5,6]. However, the frequent emergence of new variant strains complicates the control of PRV [7,8]. Therefore, there is an urgent need to develop more effective vaccines against PRV based on the new PRV strains for disease control. Yan and co-workers [9] constructed the recombinant PRV rHN1201^TK−/gE−/gI−/11k−/28k−^ strain that was derived from the HN1201 RPV variant, a virulent field strain isolated in 2011, providing a new prevention strategy for PRV. Although gene-deletion PRV vaccines provide high protection rates for preventing and controlling pseudorabies, PRV is very sensitive to temperature, and long-term exposure to ambient temperature may lead to its inactivation or reduced infectivity. Pseudorabies vaccines need to be maintained in a cold chain during transport and storage to ensure their effectiveness. Some developing countries lack the means to maintain the cold chain during transport or to establish proper storage facilities for vaccines, which is expensive [10,11]. Freeze-drying is a common technique employed in the production of live vaccines to enhance the vaccine’s biological activity [12]. Therefore, a heat-resistant vaccine for PRV is necessary to address the challenges of storing it at low temperatures and transporting it via a cold chain.

The instability of the virus is caused by water-mediated physicochemical changes and biodegradation [13]. Consequently, the removal of a significant proportion of water can impede or decelerate the virus’s instability and degradation. Freeze-drying is utilized to enhance vaccine stability. However, the process can lead to virus damage, which includes nucleic acid degradation, protein aggregation, envelope damage, and ice crystal formation [14]. To minimize the negative effects of the process, it is essential to use optimized formulations and processes. Optimization of the formulation involves determining the optimal combination of freeze-dried protectants to preserve the viral activity during lyophilization and storage. Freeze-dried protective agents, including disaccharides (trehalose and sucrose), amino acids, surfactants, and polymers, are commonly added to live vaccines [15]. Disaccharides such as sucrose, trehalose, lactose, and maltose can retain their amorphous state during freeze-drying. Additionally, they exhibit a higher T_g_ (glass transition temperature) than monosaccharides, with values of 60, 110, 114, and 100 °C, respectively [16]. Two main hypotheses, the “water substitution” hypothesis and the “glass dynamics” hypothesis, have been proposed to explain the mechanism by which protective agents exert their stabilizing effect [17,18,19]. Disaccharides tend to establish hydrogen bonds with viral capsid proteins and/or viral envelopes, thereby immobilizing surface-bound water and creating a stabilizing matrix. Upon drying, sugars replace the hydrogen bonding between biomolecules and water, as well as forming vitrified/glassy states that slow down molecular mobility and damage [20]. Animal-derived stabilizers, including gelatin and bovine albumin (BSA), are frequently utilized in the pharmaceutical sector, particularly as vaccine stabilizers. Various studies have demonstrated that gelatin interacts with vaccines via electrostatic and non-covalent interactions, ensuring vaccine stability. These additives must adhere to current guidelines [21].

In this study, heat-resistant freeze-dried formulations of the recombinant PRV rHN1201^TK−/gE−/gI−/11k−/28k−^ strain was studied. Ten different lyophilization protectants were designed, including trehalose, gelatin, thiourea, sorbitol, and polyvinylpyrrolidone (PVP), all of which have been shown to ensure the stability of live vaccines during lyophilization and storage [22,23,24]. Optimal combinations were screened using lyophilization and heat loss. The physical structure of the freeze-dried vaccine was characterized, and the mechanism by which the protective agent stabilizes the virus was elucidated. In addition, the vaccine’s long-term stability was examined, and a thermal acceleration model was established to provide a scientifically based estimate of the vaccine’s shelf life at a temperature range of 2–8 °C.

## 2. Materials and Methods

### 2.1. Materials

The recombinant PRV rHN1201^TK−/gE−/gI−/11k−/28k−^ strain was obtained from the Pulike Bio-engineering Co., Ltd. (Luoyang, China). The culture method has been described previously [9]. The viral titer of the recombinant PRV rHN1201^TK−/gE−/gI−/11k−/28k−^ virus solution was 9.0 (±0.2) TCID_50_/mL. In the experiment, fifteen 28-day-old piglets were purchased from Luoyang (China) and confirmed to be free of PRRSV, PCV2, PRV, and CSFV before being used in this study.

### 2.2. Preparation of Excipients

Ten formulations were designed using gelatin and trehalose as raw materials, and sorbitol, thiourea, and PVP as the excipients. Details of all the formulation components formulation are shown in Table 1. The ingredients of each formulation were dissolved in 20 mmol/L potassium phosphate buffer (pH 7.20–7.4). Then, filter sterilization was used for heat sensitive components (trehalose, sorbitol, thiourea and PVP), while high-pressure sterilization was used for heat-resistant components (gelatin).

### 2.3. Vaccine Lyophilization

The PRV was mixed with each formulation in a 1:1 ratio, and then 2 mL aliquots were dispensed into 5 mL sterile glass vials (Jin yuelai Pharmaceutical Glass Co., Ltd., Hebei, China). The vials were partially sealed with a butyl plug and placed on the lyophilizer (Lyo-0.5, Tofflon, Shanghai, China). The freeze-dried procedures were set as below (Figure 1): at the pre-freeze stage, samples were frozen to −45 °C within 1 h and maintained at this temperature for 3 h. The primary drying was carried out by warming to −10 °C within 1 h and maintained at this temperature for 15 h. The secondary drying occurred in two stages. Initially, the temperature was increased to 15 °C over a period of one hour and then sustained at this temperature for four hours, followed by a further increase to 25 °C over one hour, which was sustained for a further five hours. Vials were rubber-stoppered under vacuum and tightly sealed with an aluminum cap under normal air pressure.

### 2.4. Screening of Heat-Resistant Freeze-Dried Formulations

The freeze-dried vaccine was dissolved in one milliliter of DMEM, and the viral titer was measured to evaluate the freeze-dried loss. The vaccine was subsequently incubated at 37 °C for seven days to assess its heat loss. The results presented are the average of three experiments in triplicate. After comparing the freeze-dried loss and heat loss, the optimal formulation was chosen.

### 2.5. Long-Term Stability Assessment

The screened formulation was combined with PRV and freeze-dried, followed by exposure to 25, 37, and 45 °C for a long-term stability assessment. The viral titer loss was measured continuously at different time points until the viral titer reached or exceeded a loss of 1.0 Lg. The results presented are the average of three experiments in triplicate. Curve-fitting of the experimental data was performed using the Arrhenius equation [25], and the shelf life of vaccines was predicted at 4 °C.

### 2.6. Virus Titration

The freeze-dried vaccines were diluted with 1 mL of serum-free DEME cell culture medium. Further 10-fold serial dilutions (10^−3^, 10^−4^, 10^−5^, 10^−6^, 10^−7^) were made using DEME cell culture medium containing 2% fetal bovine serum (FBS). The inoculation was carried out on 96-well ST cell culture plates that had established a healthy monolayer of cells, and the nutrient solution was discarded. Eight wells were inoculated for each dilution, accompanied by eight wells of the normal cell control. The samples were then incubated at 37 °C in an incubator containing 5% CO_2_ for 5 days to observe and record cytopathic lesions. The TCID_50_ was calculated in accordance with the Reed–Muench method [22].

### 2.7. Vacuum and Moisture Content

The vaccine was repeatedly lyophilized into three batches according to the optimized formulation. The vacuum degree was controlled according to the requirements of the Pharmacopoeia of the People’s Republic of China [26]. The residual moisture content of three vaccine batches was measured via thermogravimetric analysis [27]. After drying in a hot air oven at 80 °C for 24 h, the weights of the samples were determined after calculating their dry weights.
$$Moisture\;content\;(\%)=\frac{m_{1}-m_{2}}{m_{1}}\times 100\%$$
where *m*_1_ is the initial weight, and *m*_2_ is the final constant weight.

### 2.8. Scanning Electron Microscopy (SEM)

The internal structure of the freeze-dried vaccine was investigated using scanning electron microscopy (TESCAN MIRA LMS, TESCAN, Brno, Czech Republic). The sample was sprayed with gold for 45 s and scanned at 3 kV.

### 2.9. X-ray Powder Diffraction (XRD) Analysis

The structures were characterized using an X-ray powder diffraction (SmartLab SE, Rigaku, Tokyo, Japan). Scanning angles of 5–90° 2θ were applied to the samples. The scanning speed was 2 °C/min from a wide-angle diffractometer with Cu-Kα radiation at a generator voltage of 40 kV and a generator current of 40 mA.

### 2.10. Fourier-Transformed Infrared (FTIR) Analysis

The vaccine underwent freeze-drying and was subsequently blended with potassium bromide (KBr) at a ratio of 1:100. The resulting mixture was then compressed into a pellet for the purpose of scanning (Nicolet iS20, Thermo, Waltham, MA, USA). FTIR spectra were recorded with a spectral resolution of 4 cm^−1^ over a range of 400–4000 cm^−1^.

### 2.11. Vaccine Immunogenicity and Safety

A total of fifteen healthy piglets aged 28 days were randomly assigned to three groups (n = 5). Piglets in Group 1 and Group 2 were vaccinated intramuscularly with the diluted A4 freeze-dried vaccine (5.0 Lg) and recombinant PRV rHN1201^TK−/gE−/gI−/11k−/28k−^ live virus solution (5.0 Lg), respectively. Group 3 was unvaccinated and served as a control. After vaccination, the rectal temperature and clinical signs were recorded daily. Pig serum samples at 0, 7, 14, and 21 days post-vaccination (dpv) were collected to monitor glycoprotein B (gB) and neutralizing antibody (NAb) levels. For antibody testing, gB antibodies of the serum samples were evaluated by the Aujeszky gB (Pseudorabies Virus) Antibody Test Kit (BioChek, San Francisco, CA, USA). The PRV-specific NAb titers were tested by the serum-neutralization test (SNT). Assays for gB antibody and NAbs have been described previously [9].

### 2.12. Statistic Analysis

SPSS version 25.0 (SPSS Inc., Chicago, IL, USA) was used for data analysis. The differences in the gB antibody and Nab levels in piglets between groups were determined using the Student’s *t*-test. Differences were considered statistically significant when *p* < 0.05.

## 3. Results

### 3.1. Screening of Heat-Resistant Freeze-Dried Formulations

Lyophilization consists of two steps, freezing and dehydration, both of which can lead to virus or protein denaturation. Hence, it is important to choose the right lyophilization protectant to reduce the loss of virus activity during the lyophilization process. The results are shown in Figure 2. All the gelatin-added formulations (A1–A5) were effective in stabilizing the virus during freeze-drying, with a low freeze-dried loss of no more than 0.5 Lg. Group A4 showed superior performance, with a freeze-dried loss of just 0.31 Lg. In the gelatin-free formulations (A6–A10), Group A6, which consisted of trehalose and sorbitol, exhibited the highest freeze-dried loss of 0.61 Lg. The freeze-dried loss in groups A7, A9, and A10 were lower, approximately 0.5 Lg. The lowest loss was observed in group A10, at 0.32 Lg.

The thermal stability of the A1–A10 formulations was determined by incubation at 37 °C for 7 days. The thermal acceleration test showed that A4 and A5 had the best thermal stability, with heat losses of 0.42 and 0.41 Lg, respectively. The heat loss in groups A1, A2, A6, and A7 was greater when thiourea was not added compared with the formulations that included thiourea. Virus thermal stability in the gelatin-free formulation was affected by sorbitol, resulting in a higher heat loss of 0.12 Lg for group A9 compared with group A10. Group A6, which only had trehalose and sorbitol, had the greatest loss of heat at 0.89 Lg. After considering both the lyophilization and heat losses, formulation A4 was chosen as the optimal lyophilization protectant formulation.

### 3.2. Physicochemical Characteritics of the Vaccine

#### 3.2.1. Vacuum Degree and Remaining Moisture

The freeze-dried A4 vaccine has an elegant “cake” appearance and dissolves rapidly upon dilution (Figure 3), and the sample is clear and transparent with no aggregation. The degree of vacuum pressure and residual moisture content were measured for the three batches of vaccines formulated with A4 stabilizers after freeze-drying. The results are displayed in Table 2: the vacuum degree was qualified, and the remaining moisture was less than 2%, consistent with the Veterinary Pharmacopoeia of People’s Republic of China.

#### 3.2.2. Scanning Electron Microscope

SEM was used to detect the A4-formulated vaccine’s morphology. Results are shown in Figure 4, the A4-formulated vaccine had a porous honeycomb structure, while the shape was regular.

#### 3.2.3. Fourier-Transformed Infrared (FTIR)

Hydrogen bonds are crucial in stabilizing the protein’s secondary structure, and FTIR is a widely employed reference technique for acquiring precise insights into a protein’s secondary structure. To assess whether the secondary structure of the virus-encapsulated protein was affected by storage at 37 °C, the A4 freeze-dried vaccine was placed at 37 °C for 7 days and the samples were examined for changes in their functional groups. The results showed that the absorption intensities of the absorption peaks of two characteristic bands, amide I and amide II, representing proteins and peptides, were not significantly weakened after 7 days at 37 °C, indicates that the viral proteins’ secondary structure remains intact at high temperatures (Figure 5).

#### 3.2.4. X-ray Diffraction

As shown in Figure 6, the A4 freeze-dried vaccine revealed the absence of characteristic peaks from crystalline forms in the XRD pattern. The sample tested was the amorphous material.

### 3.3. Long-Term Vaccine Stability

To further test the thermostability and storage duration of the A4-formulated vaccines, the freeze-dried vaccines were exposed at 4, 25, 37 and 45 °C and the virus titers calculated. The data presented in Table 3 indicates that at 25 °C, a heat loss of greater than 1 Lg occurred after 90 days, while at 37 °C and 45 °C, a heat loss of greater than 1 Lg occurred after 28 and 7 days, respectively.

In order to predict the shelf life of the formulations in a short time, the experimental data were fitted using the Arrhenius equation. Using linear regression of time (t) on the logarithm of the relative viral titer (lgN_0_ − lgN_t_), the loss rate equation was established to obtain the inactivation rate (K) of the A4-formulated vaccines at different temperatures (45, 37, 25 °C). As shown in Figure 7A–C, K_45_ = 0.00723 (R^2^ = 0.99801), K_37_ = 0.00158 (R^2^ = 0.98905), and K_25_ = 0.00069 (R^2^ = 0.99282). The inverse of the absolute temperature (1/T) was set as the abscissa, and the log of the inactivation-rate constant at each temperature was set as the ordinate (lgK). Based on the equation Y = −5258X + 14.305 presented in Figure 7D, the inactivation-rate constant at a temperature of 4 °C was determined to be K4 = 2.104 × 10^−5^. Subsequently, utilizing this value, the 2-year titer loss of the vaccine at 4 °C was computed to be 0.30 Lg. Additionally, the viral titer alteration at 2–8 °C was monitored under real conditions, revealing a decrease of 0.32 Lg over a period of 2 years. Ultimately, by considering a single titer decrease as the minimum storage period, the shelf life of the vaccine at 4 °C was estimated to be approximately 6.61 years.

### 3.4. Vaccine Immunogenicity and Safety

After the injections, no swelling or redness was observed at the injection sites. The body temperature, appetite, and mental state were normal in all groups throughout the study. The results showed that the A4 heat-resistant freeze-dried vaccine was safe. The immunogenicity of the freeze-dried vaccines can be evaluated by measuring gB antibody and NAb levels in the vaccinated piglets. As shown in Figure 8A, at 14 days, the serum gB antibody levels of both group 1 and group 2 reached S/P-value of 1.01 and 1.12, respectively., respectively. The serum samples were all positive. There was no significant disparity in the gB antibody levels between the two groups (*p* > 0.05). As shown in Figure 8B, the anti-PRV NAbs in pig serum elicited by group 1 and group 2 vaccination showed high levels against the HN1201 RPV variant at 14 and 21 days. Noteworthy, no gB-specific antibodies and NAbs were detected in the unvaccinated group. The above results indicate that the A4 freeze-dried recombinant PRV vaccine is equally effective as the live virus.

## 4. Discussion

The freeze-drying technique is a common method used to enhance the long-term storage stability of vaccines [28]. However, cryodamage, such as the mechanical and/or osmotic damage caused by ice crystal formation, hydrolysis, oxidation, protein denaturation and nuclear acid damage, are inevitable during the freeze-drying process [29]. For these reasons, cryodamage could affect the vaccine potency and reduce effectiveness. During the storage of vaccines, it is essential that the T_g_ is higher than the ambient temperature. If the ambient temperature of the vaccine exceeds its own T_g_, the vaccine will undergo a series of changes including collapse, hardening, and discoloration. The addition of lyoprotectants has been demonstrably effective in improving the lyophilization and storage stability of vaccines, as revealed by several studies [14,29]. In this study, we evaluated a freeze-dried protective agent for a live recombinant PRV vaccine. The thermostable protective formulation, containing 1% gelatin, 5% trehalose, 1% sorbitol, 0.5% thiourea, and 0.5% PVP, demonstrated robust stability toward the recombinant live PRV vaccine.

Among the freeze-dried protective agents, disaccharide is the most effective agent for freeze-dried vaccines [30,31]. Sugars have the ability to form a sugar matrix that has high viscosity and low mobility, thus limiting protein migration and unfolding, allowing vaccines to maintain their structural and functional integrity in the absence of water [32]. Trehalose can stabilize viruses by acting as a water replacer by forming hydrogen bonds with protein or lipid membranes. Gelatin is commonly used in freeze-dried biological products because of its ability to gel, which prevents the viral inactivation that occurs due to fluctuations in temperature. Therefore, we selected 1% gelatin + 5% trehalose and 10% trehalose as the primary protectant ingredients. The A1–A5 formulations (1% gelatin + 5% trehalose) gave better protection during lyophilization and storage than other groups, suggesting that a higher sugar concentration may hinder virus protein stabilization [33]. It is recommended to choose the lowest effective sugar concentration in the protective solution.

Trehalose possesses a high T_g_ (110 °C) and the ability to form a gel with gelatin. This may augment steric effects and affect the strength of hydrogen bonds between proteins post-lyophilization [16]. Therefore, achieving a balance between the elevated glass transition temperature and the hydrogen-bond strength of the supplementary ingredients is crucial. In the frozen state, sorbitol is traditionally deemed to be an amorphous solute that does not crystallize, and it has a low T_g_ (−1.6 °C) [34]. In the frozen state, sorbitol is typically regarded as an amorphous, non-crystalline solute. Due to sorbitol’s low T_g_ (−1.6 °C), it cannot serve as a primary ingredient in the lyoprotectant. However, it can be used as a plasticizer in lyoprotectant. In addition, sorbitol protects proteins from thermal denaturation in aqueous solution by preferential rejection. The inclusion of sorbitol in formulations containing 10% trehalose seems unfavorable in this study. There was a reduction in the lyophilization and thermal stability in groups A6–A9. In particular, in group A6, where the excipient was solely supplemented with sorbitol, the heat loss increased to 0.89 Lg. The heat loss of A10, without sorbitol, was merely 0.55 Lg in contrast to A9. However, virus stability was improved by incorporating sorbitol into gelatin-containing formulations (A1–A5), and the combination of gelatin and sorbitol has been shown to be an effective stabilizer [23,35]. While sorbitol may increase the “fluidity” of the molecule, trehalose can increase the T_g_ of the lyoprotectant. It is important to note that, at storage temperatures well below the T_g_ of the formulation, “fluidity” does not play a significant role in controlling the stability of the protein [32].

Even in a vacuum, freeze-dried vaccines can undergo oxidation during storage, and the oxidation of protein residues is a crucial degradation pathway for live vaccines [22]. Hence, thiourea was selected to be incorporated as an antioxidant in the formulation. The incorporation of thiourea did not decrease the freeze-dried loss of the vaccine but substantially diminished the heat loss of the vaccine while in storage. The A4 formulation exhibited a decrease in heat loss of 0.2 Lg compared with the A2 formulation, while A7 showcased a decrease in heat loss of 0.17 Lg compared with A9. Therefore, the incorporation of thiourea as an antioxidant in the formulation bolstered its storage stability. PVP is an amorphous polymer capable of stabilizing proteins in their dry state by increasing the T_g_. Additionally, it can prevent protein aggregation through the repulsion generated by its hydrophobic carbon [36]. Buffers are crucial in lyophilization formulations. In each formulation, 20 mM of potassium phosphate buffer was added. Potassium phosphate buffer is the most commonly used buffer because of its low pH change during lyophilization, and the pH range of the buffer (6–8) is particularly suitable for maintaining viral activity [14].

In addition to virus titration, SEM, XRD and FTIR have been widely used to evaluate heat-resistant freeze-dried formulations [24,37]. SEM results reveal that the A4-formulation vaccine exhibits a honeycomb-like porous structure, with trehalose and gelatin aiding the formation of a skeleton during freeze-drying. This structure is advantageous in facilitating the rapid sublimation of bound water during the drying stage. Additionally, the porous structure aids in the swift recovery of biological activity upon vaccine reconstitution. The protectant freeze-drying process serves a dual purpose of preserving the vaccine’s biological activity and acting as an excipient [24]. X-ray diffraction (XRD) investigations showed that the freeze-dried vaccine was an amorphous material. Amorphous formulations provide better protection than crystalline formulations. Trehalose formed an amorphous sugar–glass matrix, preventing the rupture of the virus caused by ice-crystal formation [38]. Amorphous sugars play an important role in the formation of hydrogen bonds between the virus and the protective agents during lyophilization process [38]. The FTIR results indicated that the amide I and amide II bands did not undergo significant changes after treatment at 37 °C for 7 days. The amide I band (1650 cm^−1^) corresponds to the stretching vibrations of the C=O bond in the amide molecule. The bending vibrations of the N–H bond and the stretching vibrations of the C–N bond correspond to the amide II band (1550 cm^−1^). Hydrogen bonding in the protein secondary structure involves both C=O and N–H bonds [39]. This indicates that the overall secondary structure of the protein was not disrupted. This may be mainly attributable to the replacement of water by trehalose to form a stable hydrogen bond with the protein [32].

Typically, commercial freeze-dried live vaccines require a minimum shelf life of 2 years at 2–8 °C. The most commonly used method for estimating the shelf life of biological standards with reasonable accuracy in a short period of time is the thermal acceleration test [40]. The shelf life of the samples at lower temperatures was predicted using the Arrhenius equation, which is based on the rate of rapid potency inactivation observed at several higher temperatures. Here, 25, 37, and 45 °C were selected, and the test data were fitted to a linear regression equation according to the Arrhenius equation, and then a linear regression was performed to calculate the shelf life at 4 °C. The results showed that the correlation coefficient R^2^ = 0.95 for the fitted linear regression, indicating that the choice of measurement method and time interval was reasonable. Although the method does not fully replace long-term stability studies, it enables the screening of heat-resistant formulations and scientific prediction of the shelf life of formulations for a given reasonable temperature and sampling-time design.

Prioritizing vaccine safety is crucial prior to administering any vaccine, and vaccines containing gelatin adjuvants carry a risk of allergic reactions. However, the excellent protective properties, biocompatibility, and biodegradability, and the low cost of gelatin have led to its widespread use in the lyophilization of vaccines [41,42]. gB is a significant envelope glycoprotein on the surface of the PRV [43]. It is a crucial factor in the virus’s invasion of cells and serves as a vital viral surface antigen that stimulates the production of neutralizing antibodies in the body against viral infection [44]. Therefore, gB is an ideal target for pseudorabies virus vaccine and drug design. It is often used as an antigen to monitor PRV infection in swine farms or to evaluate the immunity status of pigs in the clinic. In the present study, all piglets receiving the A4 freeze-dried vaccine were positive for gB antibodies in the serum and produced very high levels of anti-PRV NAbs. No adverse clinical signs were observed. This indicates that the A4 formulation of the freeze-dried vaccine is safe and effective.

## 5. Conclusions

Our study identified formulation A4 as an effective protective agent for the TK/gE/gI/11k/28k deleted PRV vaccine during both lyophilization and storage. The trehalose- and gelatin-based formulation stabilized the virus by incorporating it into an amorphous sugar matrix and enhancing the thermal stability of the live PRV vaccine. Furthermore, the shelf life of the formulation was accurately predicted through the utilization of a thermal acceleration test and the Arrhenius equation, thus providing a valuable reference for the expedited screening of formulations suitable for commercial vaccines. Although we screened for protectant formulations suitable for lyophilization, this study was not conducted on the process parameters of high importance, such as the freezing rate and the primary and secondary drying temperatures. No challenge study was conducted in this research. This should be considered in future studies to ensure that the antibodies produced are effective in protecting the organism from infections.

## Figures and Tables

**Figure 1 vaccines-12-00065-f001:**
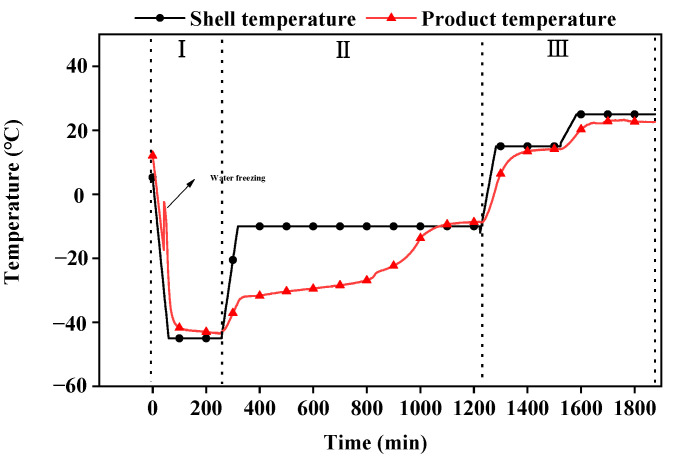
Stages of the freeze-drying procedure. I: At the pre-freeze stage, samples are frozen to −45 °C within 1 h and maintained at this temperature for 3 h. II: Primary drying is carried out by warming to −10 °C within 1 h and then maintaining this temperature for 15 h. III: The secondary drying occurs in two stages. Initially, the temperature is increased to 15 °C for one hour and sustained for four hours, followed by a further increase to 25 °C for one hour, which is then sustained for five hours.

**Figure 2 vaccines-12-00065-f002:**
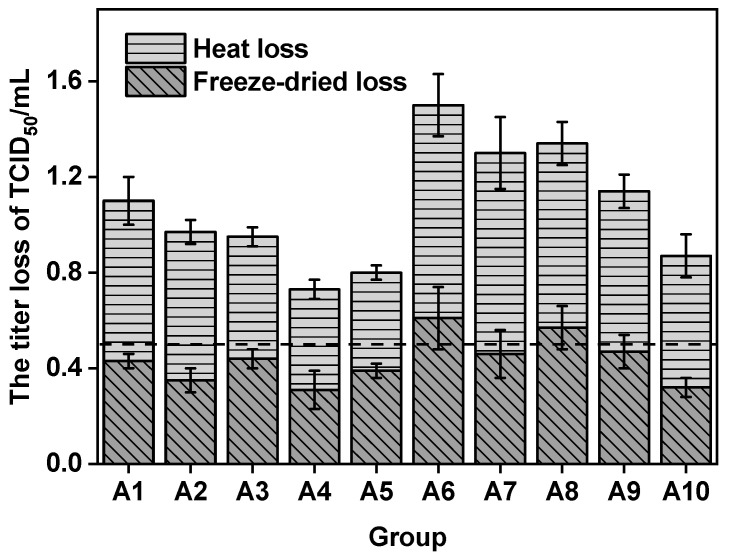
The freeze-dried loss (the difference between the vaccine titers before and after the lyophilization of the formulations) and the heat loss of the vaccines (the titer reductions for the lyophilized stabilizer formulations before and after incubating at 37 °C for 7 days), the figure displays a dashed line which represents 0.5 Lg.

**Figure 3 vaccines-12-00065-f003:**
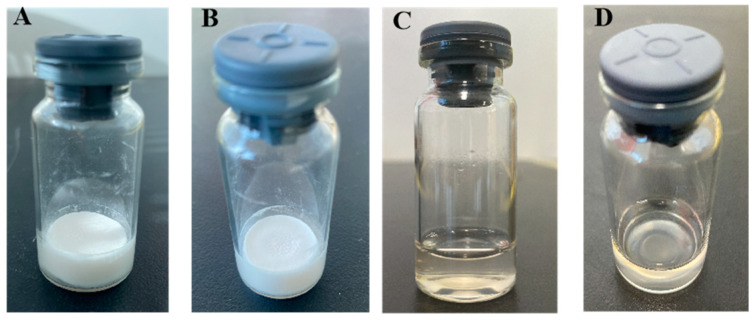
The A4 freeze-dried protectant vaccine (**A**,**B**) and the A4 freeze-dried protectant vaccine after reconstitution (**C**,**D**).

**Figure 4 vaccines-12-00065-f004:**
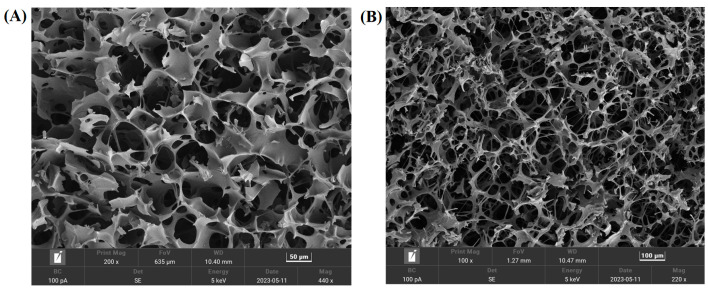
SEM of freeze-dried vaccines with formulation A4. Magnification: (**A**) ×200; (**B**) ×100.

**Figure 5 vaccines-12-00065-f005:**
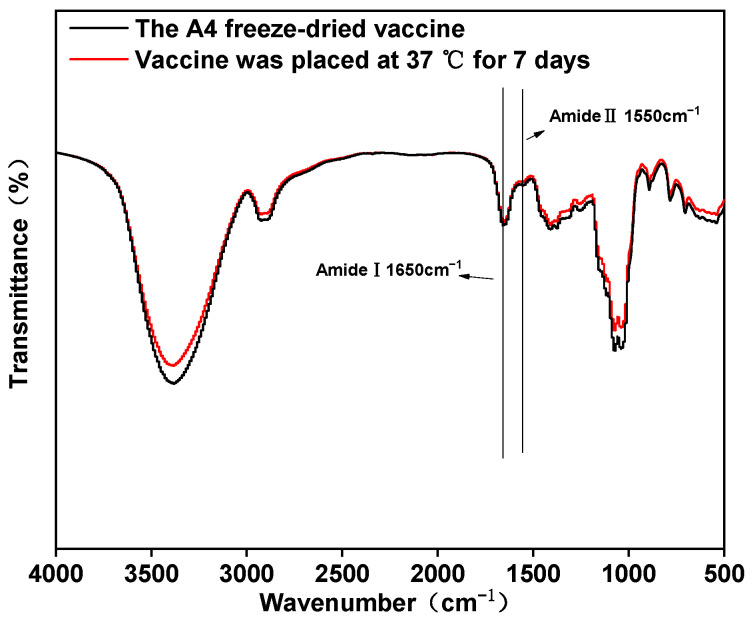
FTIR of freeze-dried powders (the amide I band corresponds to the stretching vibrations of the C=O bond in the amide molecule. The bending vibrations of the N–H bond and the stretching vibrations of the C–N bond correspond to the amide II band).

**Figure 6 vaccines-12-00065-f006:**
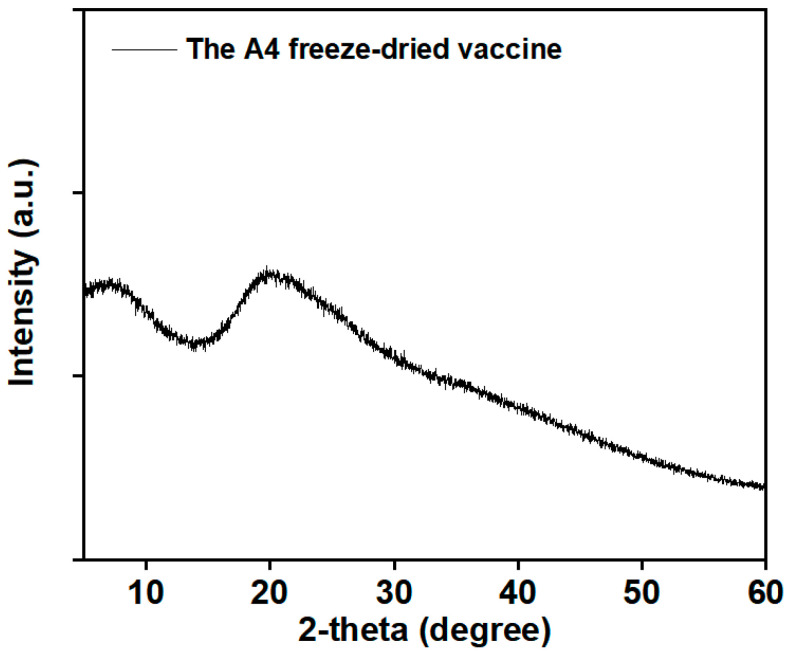
X-ray powder diffractograms of freeze-dried powders.

**Figure 7 vaccines-12-00065-f007:**
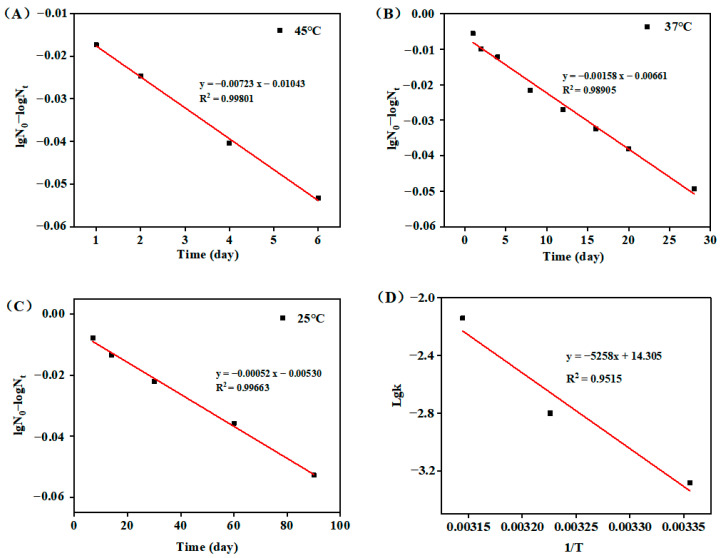
The decay rate constant K at different temperatures: (**A**) 45 °C; (**B**) 37 °C; (**C**) 25 °C. Diagram of the relationship between 1/T and the deactivation rate constant (**D**).

**Figure 8 vaccines-12-00065-f008:**
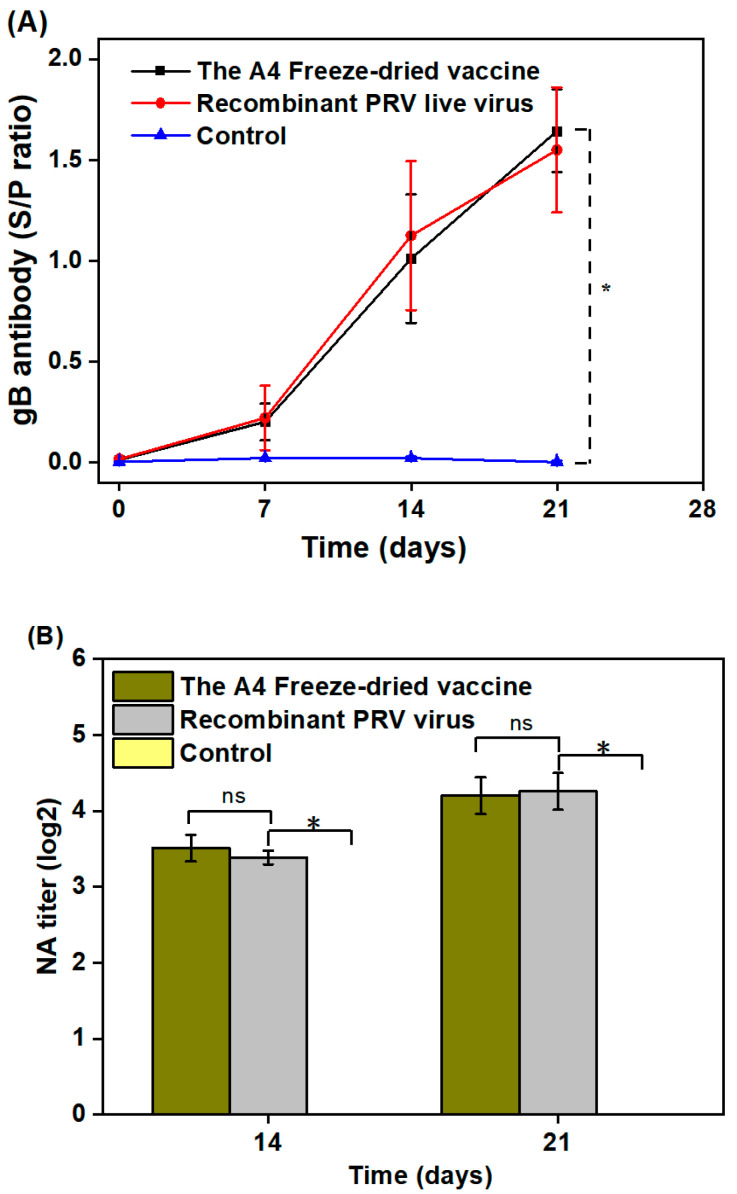
(**A**) Time-course of the PRV gB antibody response after vaccination; (**B**) NAbs against the HN1201 RPV variant at 14 and 21 days (S/P > 0.5 was considered positive; asterisk (*) denotes a statistically significant difference (*p* < 0.05); ns indicates no significant difference).

**Table 1 vaccines-12-00065-t001:** Excipient contents of PRV formulations.

Formulation	Gelatin	Trehalose	Sorbitol	Thiourea	PVP	Potassium Phosphate pH 7.0–7.4
	Final Concentration (% *w*/*w*)					mmol/L
A1	1	5	1	-	-	20
A2	1	5	1	-	0.5	20
A3	1	5	1	0.5	-	20
A4	1	5	1	0.5	0.5	20
A5	1	5	-	0.5	0.5	20
A6	-	10	1	-	-	20
A7	-	10	1	-	0.5	20
A8	-	10	1	0.5	-	20
A9	-	10	1	0.5	0.5	20
A10	-	10	-	0.5	0.5	20

**Table 2 vaccines-12-00065-t002:** Physical properties of the PRV vaccine and the remaining moisture measurement results.

Vaccine Batch Lots	The Degree of Vacuum	Remaining Moisture (%)
1	Qualified	1.7, 2.0, 1.5, 1.9
2	Qualified	1.8, 2.0, 1.7, 1.8
3	Qualified	1.6, 1.7, 1.4, 1.9

**Table 3 vaccines-12-00065-t003:** The heat loss at different temperatures.

		Temperature (°C)		
	Time (Days)	25	37	45
lgTCID_50_/mL	0	8.47 ± 0.12	8.51 ± 0.11	8.47 ± 0.15
	1	-	8.4 ± 0.08	8.33 ± 0.13
	2	-	8.32 ± 0.12	8.19 ± 0.16
	4	-	8.28 ± 0.17	7.9 ± 0.05
	7	-	8.1 ± 0.17	7.67 ± 0.21
	8	-	8.0 ± 0.23	-
	16	8.47 ± 0.13	7.9 ± 0.17	-
	20	-	7.8 ± 0.06	-
	28	8.05 ± 0.23	7.6 ± 0.21	-
	60	7.8 ± 0.14	-	-
	90	7.5 ± 0.23	-	-

## Data Availability

Data will be made available on request.

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
