# Peer review of "Screening and Stability Evaluation of Freeze-Dried Protective Agents for a Live Recombinant Pseudorabies Virus Vaccine"

_vaccines, 2024, doi:10.3390/vaccines12010065_

Round 1

Reviewer 1 Report

Comments and Suggestions for Authors

Researchers aimed to lyophilize live PRV vaccines using the known freeze-dry method. My suggestions;

1. It would be more appropriate for researchers to clearly state which substances are filter sterilized and which are high-pressure sterilized under the Preparation of excipients section.

2. Although the A4 formulation was chosen among the 10 formulations, it should be explained in the article why the A9 formulation was chosen and used in screening with SEM.

3. Some minor spelling errors in the article should be corrected. (Line 284: wate)

4. In between lines 284-292 of the discussion section, it is not fully understood why alginate is mentioned instead of Trealose and the data related to alginate are discussed, although it is not mentioned in the materials and methods section of the article, and this has caused confusion. It would be more appropriate for researchers to rearrange this part.

Reviewer 2 Report

Comments and Suggestions for Authors

The authors evaluated a number of cytoprotectant formulations to improve the long term storage of a live recombinant pseudorabies virus vaccine.

The manuscript is generally well written and easy to read there are however a number of issues that should be addressed.

1.       The authors claim that no significant differences between the A4 freeze-dried vaccine and the recombinant PRV live virus vaccines were observed with respect to gB antibody production (is this glycoprotein?, the authors should specify what exactly gB is). No indication of the statistics used or the power of the test was provided, this should be included.

2.       The authors give no indication of the physical condition of the vaccine post reconstitution, i.e. is there aggregation of the samples?

3.       The authors should provide more detailed figure legends.

4.       To provide a more sound argument the authors could demonstrate that their A4 freeze dried formulation provides similar protection and the recombinant PRV live virus vaccine in a challenge study.

Round 2

Reviewer 2 Report

Comments and Suggestions for Authors

Reviewers comments have been answered.